# The Relationship between Autism and Ehlers-Danlos Syndromes/Hypermobility Spectrum Disorders

**DOI:** 10.3390/jpm10040260

**Published:** 2020-12-01

**Authors:** Emily L. Casanova, Carolina Baeza-Velasco, Caroline B. Buchanan, Manuel F. Casanova

**Affiliations:** 1School of Medicine Greenville, University of South Carolina, Greenville, SC 29615, USA; m0casa02@louisville.edu; 2Laboratory of Psychopathology and Health Processes, University of Paris, 92100 Boulogne Billancourt, France; carolina.baeza-velasco@u-paris.fr; 3Department of Emergency Psychiatry and Acute Care, CHU Montpellier, 34000 Montpellier, France; 4Greenwood Genetic Center, Greenville, SC 29605, USA; cbuchanan@ggc.org; 5Department of Psychiatry and Behavioral Sciences, University of Louisville, Louisville, KY 40292, USA

**Keywords:** autism spectrum disorder, Ehlers-Danlos syndrome, hypermobility spectrum disorders, autonomic disorder, mast cell activation syndrome

## Abstract

Considerable interest has arisen concerning the relationship between hereditary connective tissue disorders such as the Ehlers-Danlos syndromes (EDS)/hypermobility spectrum disorders (HSD) and autism, both in terms of their comorbidity as well as co-occurrence within the same families. This paper reviews our current state of knowledge, as well as highlighting unanswered questions concerning this remarkable patient group, which we hope will attract further scientific interest in coming years. In particular, patients themselves are demanding more research into this growing area of interest, although science has been slow to answer that call. Here, we address the overlap between these two spectrum conditions, including neurobehavioral, psychiatric, and neurological commonalities, shared peripheral neuropathies and neuropathologies, and similar autonomic and immune dysregulation. Together, these data highlight the potential relatedness of these two conditions and suggest that EDS/HSD may represent a subtype of autism.

## 1. Introduction

Autism is a complex spectrum condition. Most of the autistic population is considered “idiopathic” with causes unknown, likely the result of complex polygenic and environmental interactions [1,2,3,4]. Meanwhile, a substantial minority on the autism spectrum display rare genetic variants that appear to be the primary cause of their conditions [5]. Often the autistic phenotype associated with these rare variants is secondary to a genetic syndrome (aka, syndromic autism) and is accompanied by intellectual disability and other physical impairments such as multiple congenital anomalies [6]. Popular examples of syndromic autism include fragile X syndrome (FXS) (1–5:10,000) and tuberous sclerosis (TSC) (1–5:10,000), but also include even less well known and even rarer syndromes such as Lowe syndrome (OCRL) (1:500,000) and mucopolysaccharidosis type 3 (MPS3) (1–9:1,000,000) [7,8]. To date, there are more than 60 monogenic syndromes with high penetrance for autism, as well as other forms of syndromic autism that are the result of larger chromosomal abnormalities [6,9,10]. The severity of the autism phenotype varies across the entire spectrum, although individuals with rare (often de novo) deleterious gene variants tend to be more severely affected; meanwhile, individuals of average or above-average cognitive ability tend to harbor a higher polygenic load of small effect variants, which are often inherited and may also be linked with the broader autism phenotype (BAP) in parents and siblings (reviewed in [11,12]).

Like autism, Ehlers-Danlos syndromes (EDS)/hypermobility spectrum disorders (HSD) appear to be complex spectrum conditions and are classed as hereditary connective tissue disorders (HCTD). According to the International EDS Consortium [13], there are currently 14 recognized subtypes of EDS, 13 of which are considered “rare,” each occurring in no more than 1:2000 individuals and usually far rarer. All are associated with rare gene variants, often targeting collagen pathway genes [13,14]. Meanwhile, although there are currently no accurate prevalence rates or known genetic associations for the remaining subtype, hypermobile EDS (hEDS), clinical opinion and the fact that it makes up 80–90% of EDS cases strongly suggests it is a common condition. It is therefore probable the majority of hEDS cases are associated with small effect polygenic risk factors and environmental exigencies [15,16]. A whole genome/exome sequencing investigation, the Hypermobile Ehlers-Danlos Genetic Evaluation (HEDGE) study, is currently underway that will hopefully help to address some of these questions [17].

Previous research indicates that the new diagnostic entity known as generalized hypermobility spectrum disorder (G-HSD) (which partly takes the place of joint hypermobility syndrome or JHS) occurs in roughly 0.75–2% of the population and is defined by generalized joint hypermobility and chronic musculoskeletal pain and/or instability [15]. Many individuals with a current diagnosis of G-HSD could have previously received an EDS diagnosis, but since stricter changes to the nosology in 2017 patients must now meet additional criteria involving features such as the skin, hernias/prolapses, Marfanoid habitus, and heart malformations [15]. There is considerable ongoing debate amongst the patient and medical communities as to whether these additional clinical signs truly delineate two unique entities (hEDS vs. G-HSD) or are an arbitrary line drawn in the diagnostic sand. Research seems to indicate that the two conditions blend into one another and the diagnoses are poor predictors of overall physical impairment and prognosis, suggesting criteria may well change again in future [18,19,20].

## 2. Autism and Ehlers-Danlos Syndrome Comorbidity and Familial Co-Occurrence

There is a small but growing body of literature highlighting the overlap between autism and EDS/HSD. Early research includes mainly case studies [21,22,23], but later research has provided population-level evidence of a relationship [24]. There has also been a number of studies investigating joint hypermobility (irrespective of HCTD) and its relationship to neurodevelopmental conditions such as autism and ADHD [25,26,27,28].

Unfortunately, because these two spectrum conditions tend to be diagnosed and treated by different clinical professionals, it is not often that their comorbidity is recognized, likely leading to significant underdiagnosis [28]. For instance, although developmental-behavioral pediatricians may test for hypotonia and joint laxity in autistic children during initial assessment, unless signs are severe and suggestive of an underlying genetic disorder they are usually ascribed to the autism itself, a result of diagnostic overshadowing.

In addition, it is well-recognized that females with autism are an underdiagnosed population, particularly those who fall within the intellectually-abled end of the spectrum as they may have a different symptom presentation, are usually more skilled at social masking or rehearsed mimickry, and may experience different psychiatric comorbidities than their male counterparts [29,30,31]. Because hEDS is overwhelmingly diagnosed in women, it is therefore likely that autism spectrum conditions are underrecognized in this clinical subpopulation [32].

Preliminary work from Casanova et al. [16] also suggests that autism and EDS/HSD co-occur within the same families. The researchers found that more than 20% of mothers with EDS/HSD reported having autistic children—a rate not significantly different from those reported by mothers who themselves are on the autism spectrum. In addition, the rates of autism in the children shared a significant positive relationship with the severity of maternal immune disorders in EDS/HSD, suggesting that the mother’s immune system may play an additional role in autism susceptibility in these connective tissue disorders. Interestingly, maternal immune activation (MIA) appears to play a significant role in many cases of idiopathic autism, suggesting a shared mechanism of risk [33]. We will discuss these relationships in greater detail later in the manuscript.

## 3. The Genetics of Hypermobility

The majority of gene mutations associated with the rarer forms of EDS involves fibrillar collagens, proteins that modify collagen, or enzymes involved in collagen processing [13]. Interestingly, one type of EDS, known as the periodontal type, is associated with variants in two different complement genes involved in the innate immune system, leading to chronic activation of the complement system independent of microbial triggers [34]. Although *C1R* and *C1S* are not collagen-related genes per se, synthesis of types I and III procollagen is nevertheless impaired in this form of EDS, suggesting upstream effects [35].

Recently, Tassanakijpanich et al. [36] (unpublished data) have collected a series of case studies identifying a fully presenting hEDS phenotype in adult female fragile X premutation carriers, including a case with Marfanoid habitus, arachnodactyly, and right ventricular dilation. Previous reports have linked hypermobility with fragile X syndrome (FXS) and fragile X-associated disorders, although most studies have focused on hypermobility within the distal small joints and have not addressed generalized hypermobility as often [37]. (As a note, most genetics centers do not include the Beighton assessment for generalized joint hypermobility within their fragile X protocols, which may help explain why some aspects of joint hypermobility are overlooked.)

Although data are still preliminary, the presence of an EDS-like phenotype associated with the *FMR1* gene, a negative regulator of protein translation, suggests upregulated protein translation may lead to collagen dysregulation. For instance, matrix metallopeptidase 9 (MMP9) levels are disturbed in FXS as a direct consequence of low or absent Fragile X Mental Retardation Protein (FMRP) [38]. MMP9 is a known regulator of various types of fibrillar collagen, suggesting that collagen synthesis and/or secretion may be altered in FXS and fragile X-associated disorders [39]. Because FMRP negatively regulates a wide variety of proteins, it is likely that it has other indirect effects on collagen synthesis and modification in addition to MMP9. Interestingly, Rett syndrome, which is associated with *MECP2* deletions, is also associated with joint hypermobility [40]. The MECP2 protein, like FMRP, is a major (positive) regulator of collagen deposition and Rett syndrome fibroblasts exhibit a notable reduction in collagen I synthesis, once again linking upstream dysregulation of collagen to the hypermobile phenotype [41,42].

It is intriguing, and perhaps unsurprising in the context of this paper, that both of the above-mentioned syndromes share strong ties with autism. In fact, there is a substantial list of syndromic forms of autism that share a hypermobile phenotype. We have collected 35 such monogenic syndromes from the Online Mendelian Inheritance in Man (OMIM) [43] database with strong associations to autism and hypermobility (OMIM search terms: “joint (hypermobility OR hyperlaxity OR laxity OR hyperextensibility) AND autism”) (Table 1). Syndromes that had tenuous associations with either autism or hypermobility, were extremely low in patient numbers, or involved multigene deletions/duplications were removed (see Appendix A). We likewise collected the 12 different rare subtypes of EDS with their 19, respectively, associated genes into a similar list (hEDS is absent due to lack of gene associations) (Table 1).

We took the 35 autism/hypermobility (A–H) genes, together with the 19 EDS genes, and ran them through GeneMANIA to produce an extended gene interaction network based on direct physical interactions between proteins, shared pathway involvement, and genetic interactions [44] (see Appendix A). As can be seen in Figure 1, the EDS and A–H genes cluster extensively, suggesting substantial interactions between these gene networks and a potential mechanism for phenotypic overlap between autism and hypermobility-related disorders.

These data indicate that, in future, our genetic concept of collagen-related disorders may expand significantly beyond the Ehlers-Danlos syndromes as we currently know them. Undoubtedly, their heterogeneity may become more apparent as whole genome sequencing studies continue to be performed.

## 4. Symptom Overlap between Autism and Ehlers-Danlos Syndromes/Hypermobility Spectrum Disorders

Although the criteria of these two complex spectrum conditions appear different on paper (one is defined by neurobehavioral symptomology, while the other is defined by structural manifestations of connective tissue impairment), they share not only comorbidity and familial co-occurrence but symptom overlap. In this section, we will review these similarities.

### 4.1. The Nervous System

#### 4.1.1. Neurobehavioral, Psychiatric, and Neurological Features

It is well known that high rates of comorbidity exist between various neurodevelopmental conditions (e.g., autism, attention-deficit/hyperactivity disorder (ADHD), learning disorders, and motor disorders), which is more the norm than the exception. Indeed, up to 78% of children with autism present with comorbid ADHD [45]. In addition, autism is more frequent among people with learning disabilities and these share an inverse relationship with IQ [46]. As we see in autism, neurodevelopmental comorbidities also frequently co-occur in EDS/HSD [47,48]. In a study involving a large cohort of patients with EDS (N = 1771) and JHS (N = 10,019), the researchers observed an increased risk for ADHD in both samples, as well as in unaffected siblings [24]. Later, Piedimonte et al. [49], who explored developmental attributes in a group of 23 children with different HCTD (22 with EDS/HSD and one with Loeys-Dietz syndrome), observed that 61% presented with some kind of comorbid neurodevelopmental disorder (26% developmental coordination disorder (DCD); 22% learning disorder, 9% ADHD, and 4% ADHD plus DCD). Tourette syndrome, which also co-occurs with autism [50], has recently been associated with generalized joint laxity, orthostatic intolerance, and pain [26], as well as other neurodevelopmental conditions such as ADHD and autism. In addition, Adib et al. [51], Ghibellini et al. [52] and Piedimonte et al. [49] highlighted the presence of learning disorders in children with EDS/HSD.

Neurodevelopmental issues in EDS/HSD may also be related to proprioceptive impairment, which alters coordination and posture, and likewise may be involved in the acquisition of verbal communication and motor competence [52]. Baeza-Velasco et al. [28,48] have proposed that in order to maintain motor competence despite proprioceptive impairment, executive function may be overwhelmed in those affected, leading to some symptoms reminiscent of ADHD. In addition, pain and dysautonomia, which are frequently experienced by people with EDS/HSD, have also been associated with cognitive deficits in attention and concentration [53,54]. Thus, certain features present in EDSH/HSD, such as hypermobility, dysautonomia, chronic pain, and proprioceptive impairment, may have consequences in terms of motor, cognitive, and behavioral skills, and may ultimately affect aspects of neurodevelopment [28,48,55]. These hypotheses require further scientific exploration.

Other psychiatric features that are common in both autism and EDS/HSD include: anxiety, depression, bipolar disorder, eating disorders, and suicidal behaviors [24,56,57,58,59,60,61,62]. In particular, anxiety and mood disorders share strong links with autonomic dysregulation (which we will discuss in further sections) and may be related to sympathetic hyperarousal/parasympathetic hypoarousal and the chronic fatigue that often results [63,64].

Certain neurological conditions are also more common in both EDS/HSD and autism. In autism, for instance, the lifetime risk for developing epilepsy ranges between 2.7% to 44.4%, which is a seven-fold increased risk compared to the general population [65]. Individuals with intellectual disability experience the highest rates of epilepsy at approximately 22%; however, autistic people without intellectual disability still develop epilepsy at a rate of about 8% as compared to 0.75–1.1% within the general population [65,66,67]. In addition, abnormal electroencephalograms (EEG), even in the absence of epilepsy, have been reported in up to 60% of those with autism [66].

Seizure disorders have also been reported in EDS/HSD, particularly in association with certain subtypes such as an EDS-like disorder associated with mutations in the *FLNA* gene [68]. This form of EDS often presents with periventricular heterotopias, a structural anomaly that has also been noted in autism [69,70]. Raw data from our own previous study [16] likewise indicate that, compared to sex-matched controls, women with EDS/HSD report higher rates of epilepsy (5% vs. 1%), a figure that is similar to those with autism without intellectually disability (Mann–Whitney U, one-tailed, *W* = 17284.5, *p* = 0.042, EDS/HSD N = 367, control N = 98). (See Appendix A for abbreviated raw data). It should be noted, however, that some individuals within our study—though not all—reported seizure disorders following some form of head trauma [16]. Higher rates of head trauma in this clinical population have been reported in our study and others [71], possibly related to events of syncope or even coordination issues and increased numbers of accidents. Interestingly, a longitudinal study utilizing data from the Taiwanese National Health Insurance Research Database reported links between traumatic brain injury (TBI) and various neurodevelopmental conditions, including autism [72], suggesting TBI-related epilepsy may be underappreciated in these neurodevelopmental conditions.

Like epilepsy, sleep disorders are also overrepresented in both of these clinical populations. Within autism, although sleep problems persist across age groups, the types of sleep problems may vary. In younger children, bedtime resistance, sleep-related anxiety, night wakings, and parasomnias appear to be the most common issues. Meanwhile, older children and adolescents are more likely to experience delays in sleep onset, decreased sleep duration, and sleepiness during the daytime hours [73,74].

In EDS/HSD, insomnia, hypersomnia, and periodic limb movement disorder are often present; however, this population also tends to have problems sleep-disordered breathing (SDB), including obstructive sleep apnea (OSA) [75,76]. In particular, individuals with OSA report excessive fatigue and daytime sleepiness, which can be helped by nasal continuous positive airway pressure (CPAP) [76,77]. By comparison, although SDB has been little studied in autism, one report by Elrod et al. [78] indicated that individuals with autism had higher rates of SDB as compared to controls. These findings suggest that—like TBI—apneas, hypopneas, and other insufficiencies in ventilation may warrant further investigation in the autism population.

#### 4.1.2. Coordination Problems and Sensory Issues

As briefly discussed in the previous section, motor coordination is significantly impaired in autism and may be apparent from an early age [79]. Delays in motor milestones have been consistently reported by parents, as well as noted in experimental studies (reviewed in [80]). In addition, these deficits may become more apparent with age and lead to notable developmental coordination concerns in the majority of children on the spectrum, including females [81,82,83].

Children with developmental coordination disorder (DCD) (without a diagnosis of autism) experience significant proprioceptive impairment, which appears to be at the root of the condition [84,85,86]. Some studies have reported a similar proprioceptive impairment in autism, suggesting links with the coordination issues observed [87,88]. Like autism, EDS/HSD also frequently presents with coordination and proprioceptive deficits and may lead to further complications and injury in these HCTD [52,89,90]. Given the neurodevelopmental profile gradually being recognized in EDS/HSD, it is possible that issues with motor coordination and proprioception arise not only from connective tissue dysfunction but via a neurodevelopmental component as well [52].

In the context of specialized clinics for connective tissue disorders, a clear relationship between generalized joint hypermobility and a characteristic neurodevelopmental profile affecting coordination is emerging. The clinical features of these patients tend to overlap with those of developmental coordination disorder and can be associated with learning and other disabilities. Physical and psychological consequences of these additional difficulties add to the chief manifestations of the pre-existing connective tissue disorder, affecting the well-being and development of children and their families [52].

People with autism also typically experience sensory issues in the form of both hyper- and hypo-sensitivities, which depend on the individual, the organ involved, and even the surrounding environment. For instance, the sense of touch is often affected, although the presentation is variable. The majority of intellectually-abled (IQ > 70) autistic individuals tend to experience increased pain and touch sensitivity, particularly in areas innervated by small unmyelinated C-fibers [88]. Chien et al. [91] reported the presence of small C-fiber pathology (denervation) in more than half of their adult male cases, a finding that shared a U-shaped relationship with autism severity. Another small study by Silva and Schalock [92] investigated C-fiber innervation in the skin of four autistic children with hypoesthesia and allodynia, again finding reduced density of nerve fibers within the regions studied. Notably, small fiber neuropathy often leads to hyperalgesia and allodynia, due in part to hyperreactivity of the affected nerves to typical pain-modulating effectors like substance P [93]. Hypoesthesia is also common in conjunction with hyperresponsivity to non-painful or mildly painful stimuli.

Remarkably similar to the findings in autism, the vast majority of EDS/HSD individuals—including those with some of the rarer forms of EDS—experience neuropathic pain including generalized hyperalgesia and exhibit denervation of C-fibers upon skin biopsy [94,95,96,97]. The exact same features are found in a mouse model of classic EDS (cEDS) (*Col5a1^+/−^*), including allodynia and hyperalgesia, as well as the denervation of small C-fibers seen in the human condition [98]. Although the mechanism is not well understood, it is clear that connective tissue dysfunction lies upstream of such peripheral neuropathies. In contrast, most sensory disturbances in autism have been presumed to derive from pathology of the central nervous system; however, the data reviewed here suggest that peripheral nerves may also be involved.

Autistic people also experience sensory sensitivities in other modalities, such as light and sound hypersensitivity [99]. Although these modalities have been little studied within EDS/HSD specifically, they do share overlap with a common comorbid condition known as postural orthostatic tachycardia syndrome (POTS), which often involves light and sound sensitivity as part of its neurological sequelae [100]. Interestingly, cardiac function (especially baroreflex activity), which is abnormal in POTS, has been shown to have a significant effect on the processing of sensory information in the brain, suggesting links between this form of dysautonomia and sensory abnormalities [101,102]. In the next section, we will also discuss a variety of neurological and spinal stability issues that are common to EDS/HSD that predispose towards sensory processing abnormalities and that, in some instances such as Chiari I malformation, share overlap with autism [103,104].

#### 4.1.3. Autonomic Dysregulation

There is a large body of literature describing autonomic dysregulation in autism. The profile in autism is typified by high basal sympathetic (fight or flight) tone, lower parasympathetic (rest and digest) activation, and low sympathetic reactivity to certain stimuli including tests of orthostatic tolerance [105,106,107,108]. However, a small percentage of more severely affected cases tend to have sympathetic under activation except when engaging in self-injurious behaviors, which suggests these behaviors are a form of autonomic self-regulation. Similarly, in others on the spectrum with sympathetic hyperarousal, stimulatory behaviors are a means of self-calming [107].

Sympathetic hyperarousal can be inferred in this population from higher basal heart rate, increased pupillary size, and higher respiration rate, all of which have been regularly reported in autism [105,109,110]. Autistic people also have higher sympathetic tone during certain stages of sleep (N2, N3, REM) suggesting potential sleep disturbance, and during social interactions with peers [111,112]. Interestingly, when pets are included in these peer interactions, children with autism exhibit comparatively lower sympathetic arousal, suggesting animal therapy may be a useful technique in mediating autonomic dysregulation [112].

As mentioned, autistic people tend to have lower parasympathetic tone, which is most exaggerated during the morning hours [113]. In addition, usually the lower the parasympathetic tone the greater the symptoms of anxiety and the poorer the social skills [63,114]. Children with autism who have lower tone also tend to have more lower gastrointestinal problems such as constipation, which are particularly common in children who experience regression or skills loss [115].

Despite the breadth of literature on autonomic dysregulation in autism, the available treatments are comparatively few. The beta blocker, propranolol, has been used with some modest success in order to treat various symptoms in autism, with positive effects on autonomic symptoms, although randomized controlled trials are still needed [116]. Unfortunately, its use is contraindicated in individuals with asthma, which is a condition that may share greater comorbidity with dysautonomias in pediatric populations [117]. Our group has also had significant success treating dysautonomia in autism using low-frequency repetitive transcranial magnetic stimulation (rTMS), with the hopes of expanding its use to other autonomic disorders in future [118,119].

As with autism, dysautonomias are common extraarticular manifestations in EDS, particularly in the hypermobile type, and they significantly influence quality of life for these individuals. According to one study, the extent of autonomic burden in hEDS is similar to that seen in fibromyalgia, both of which are comparatively more severe than that found in classic and vascular types of EDS [120]. Similar to the profile in autism, hEDS patients typically exhibit high resting sympathetic tone but blunted sympathetic reactivity to certain stimuli (e.g., valsalva maneuver, tilt test) [120,121]. Some individuals with these conditions, particularly those with POTS (mentioned in the previous section), also seem to exhibit a weakened baroreflex response due to dysregulated vagal (parasympathetic) efferent activity [102].

Clinical manifestations of these autonomic disorders include tachycardia, hypotension, gastrointestinal disorders (particularly those relating to motility), bladder dysfunction (e.g., urinary frequency), and poor temperature regulation. Although the dysautonomia present in autism does not typically receive a diagnostic label and is usually only symptomatically described, people with EDS frequently receive comorbid diagnoses of POTS, orthostatic hypotension (OH), mixed POTS/OH, orthostatic intolerance (OI), and neurally mediated hypotension (NMH) [122].

The first line of interventions for these types of autonomic disorders, particularly in those patients who are not experiencing significant disability, are primarily behavioral and focus on avoiding triggering factors (e.g., avoid standing for long periods of time in order to reduce blood pooling within the legs), increasing fluid intake (particularly with sodium to increase blood volume), taking salt tablets or adding more salt to the diet, wearing compression garments on the lower limbs, and performing regular cardiovascular exercise [122]. Unfortunately, cardiovascular exercise may be a challenge for some individuals with EDS/HSD due to connective tissue instability/inflammation, leading to further injury, thus placing some patients with comorbid autonomic disorders in a “catch 22”.

Pharmacological treatments, which are often used in moderate-to-severely affected individuals, include medications such as the corticosteroid, fludrocortisone; the alpha-adrenergic agonist, midodrine and various beta blockers. Interestingly, for those individuals who also have endocrine disorders such as dysmenorrhea, menorrhagia, or menstrual irregularity, hormonal contraceptives can also be a first line of treatment for dysautonomias. In addition, intravenous normal saline is often helpful for individuals experiencing acute episodes or as weekly infusions for those intolerant of other treatment approaches [122].

There are a variety of neurological and spinal manifestations common to EDS/HSD that seem to predispose towards or worsen autonomic symptoms, particularly those that are related to instability and deformation of the brain stem/spinal cord. These include conditions such as Chiari I malformation, idiopathic intracranial hypertension, tethered cord, cerebrospinal fluid leaks, cranio-cervical instability, and atlantoaxial instability. In general, for those conditions in which surgical intervention is an option, surgery tends to significantly improve autonomic disorder severity [103].

Interestingly, research by Jayarao et al. [104] suggests that approximately 7% of autistic children who had an MRI for any reason exhibited evidence of Chiari I malformations, with about half of those children asymptomatic. Given, however, that most MRIs are performed in the supine rather than upright position, Chiari malformations may have been missed in some cases. Therefore, this series may be an underestimate—although the sample was likely biased considering the collection method (individuals prescribed MRI) [123]. The presence of Chiari in a significant minority of autistic patients, however, suggests the possibility of heritable disorders of connective tissue.

As discussed in the previous section, many autistic people display signs of small fiber neuropathy within the skin [91,92]. These same types of small unmyelinated C-fibers are intimately involved in the autonomic nervous system, suggesting that structural and functional damage may also be present within this system as well. Similarly, the vast majority of EDS patients exhibit small fiber neuropathy on skin biopsy, once again drawing potential links between the neuropathy within the somatosensory systems in EDS/HSD and autism and the autonomic disorders they experience [95].

### 4.2. Immune Dysregulation

The literature exploring immune dysregulation in autism is broad, ranging from upregulated pro-inflammatory cytokine levels in cerebrospinal fluid and blood, brain-specific autoantibodies (both in patients and mothers), and changes to immune cell function (e.g., downregulated T regulatory cells) [124,125]. The influence of the maternal immune system has also been explored in the form of human studies, cell lines, and maternal immune activation (MIA) animal models, illustrating the direct influences of maternal immunomodulators and autoantibodies on embryonic and fetal brain development (reviewed in [33]). Jones et al. [126], for example, found that mothers of children with both autism and intellectual disability (ID) produced significantly more cytokines and chemokines during pregnancy than mothers of autistic children without ID, indicating these immunomodulators may be playing an additional role in the severity of the children’s conditions.

Cytokines and chemokines are known to derail preprogrammed brain development by altering cell proliferation, differentiation, and dendrite and synapse formation. For example, Smith et al. [127] found that maternal exposure during pregnancy to the immunostimulant, polyI:C (a toll-like receptor 3 agonist), resulted in thickening of the developing neocortex in the offspring—with the thickness of different layers varying according to the timing of exposure (i.e., earlier exposures tended to affect the lower layers, which develop earlier, while later exposures influenced the upper neocortical layers). Similarly, Gallagher et al. [128] found that maternal exposure to the inflammatory cytokine, interleukin 6 (IL-6), resulted in significant expansion of the forebrain neural progenitor pool of the offspring well into adulthood, suggesting permanent changes occurred affecting both the forebrain cell population and its epigenome. Cortical thickening has often been noted in autism on MRI [129,130].

Although EDS/HSD are typically thought of as collagen-based disorders, like autism there is evidence for significant immune involvement in these conditions. As discussed earlier, one type of EDS, known as the periodontal type, is the result of mutations in immune-mediating genes (*C1R*, *C1S*) that help regulate activation of the complement system. Mutations in these genes lead to chronic overactivation of this system, but further downstream also lead to reduced synthesis of procollagens I and III and ultimately the classic symptoms of EDS. Periodontal EDS strongly suggests that the immune system is an important mediator of connective tissue synthesis and therefore certain immune disorders may predispose towards the Ehlers-Danlos syndrome.

On the other hand, it is also possible that chronic collagen dysregulation and subsequent tissue injury may lead to chronic immune dysregulation, as evidenced by the mast cell-related disorders that are such a common comorbid feature in EDS/HSD [131]. For instance, one preliminary study by Chang and Vadas [132] found that all of the patients they studied with hEDS and/or POTS displayed symptoms indicative of mast cell activation, including cutaneous, gastrointestinal, naso-ocular, cardiovascular, respiratory, and central nervous system symptoms. In addition, the vast majority of these patients responded well to treatment with histamine receptor 1/2 blockers and mast cell stabilizers.

In support of this relationship, particularly within the hEDS phenotype, Chiarelli et al. [133] reported that while fibroblasts from classic and vascular EDS patients displayed perturbed collagen biosynthesis and processing, hEDS fibroblasts were typified by a pro-inflammatory myofibroblast-like state, suggesting a process of chronic abnormal wound repair. Therefore, the originating cause in many cases of hEDS may be upstream of collagen synthesis and mediated by the immune system. Both the immune system and connective tissue engage in significant crosstalk and it is therefore possible that impairment may arise anywhere along this reversible feedback loop, leading to a common phenotype [16,134,135]. For instance, in utilizing raw data reported originally in Casanova et al. [16], we show statistically similar profiles of immune-mediated symptoms as reported by female patients across hEDS, cEDS, and vascular EDS (vEDS) subgroups, suggesting that—at least in the case of cEDS and vED—primary collagen impairment also leads to immune dysregulation (Mann–Whitney U, 2-tailed, *W* = 239–4960.5, *p* = 0.080–0.232) (Figure 2). (See Appendix A “Tab7_EDS_Immune” for abbreviated dataset; see [16], Appendix A for full raw datasets). Of relevance to the relationship between hEDS and fragile X premutation discussed earlier, immune-mediated disorders have also been reported in carriers, a finding linked with *FMR1* CGG repeat number [136,137,138].

As mentioned previously, Casanova et al. [16] also found that EDS/HSD mothers reported having unusually high rates of autistic children and that these EDS/HSD mothers tended to report more immune-mediated symptoms. This suggests that the maternal immune system may have played a role in autism susceptibility in these families. This work, however, is still preliminary and requires clinical studies to confirm these associations.

## 5. Conclusions

Although autism is defined neurobehaviorally and EDS/HSD by various articular and extra-articular connective tissue manifestations, these two conditions share considerable phenotypic overlap at various levels. Genetic data indicate similarities at the molecular, cellular, and tissue levels, as illustrated by numerous genetic syndromes with comorbid autism and hypermobility, which we have reviewed within this manuscript. EDS/HSD and autism comorbidity and familial co-occurrence lend further credence to this relationship, suggesting potential links via the maternal immune system

Meanwhile, these two spectrum conditions share a variety of secondary comorbidities, including similar neurobehavioral, psychiatric, and neurological phenotypes, such as ADHD, anxiety and mood disorders, proprioceptive impairment, sensory hyper-/hyposensitivities, eating disorders, suicidality, epilepsy, structural abnormalities such as Chiari I malformation and periventricular heterotopias, and sleep disorders—particularly those involving SDB. Relevant to these neurophenotypes are also common autonomic disorders (sympathetic hyperarousal, low parasympathetic tone) and immune disorders, which may influence cognition (e.g., anxiety, depression, fatigue, sleep disorders).

In consideration of the materials presented in this review, we (along with previous authors) have proposed that hereditary connective tissue disorders represent a subtype of autism whose prevalence is currently unknown, although the common nature of HSDs (and likely hEDS) suggests it may comprise a significant minority of autism cases. This relationship indicates that connective tissue impairment may influence brain development, either through direct and/or indirect means. Future studies will ideally involve in vitro and in vivo research to address these possibilities and further define the causative factors in autism susceptibility.

### Precision Medicine and HCTD in Autism

Given the push towards personalized and precision medicine in clinical practice, we strongly recommend that the Beighton scoring system be integrated into the standard physical assessment for those individuals on or suspected to be on the autism spectrum [15,139]. Training is required in the use of this simple assessment which may be taught by physical therapists, geneticists, or rheumatologists, and we strongly recommend the use of a goniometer for accuracy and the use of important landmarks for measurement. Gauging hypermobility by eye, particularly for the larger joints like the knees and elbows, can easily be underestimated for those individuals falling within the 11–13° range, leading to false negatives and missed diagnoses. Therefore, the use of the goniometer and repeated measurements are highly recommended. For those individuals who meet criteria for generalized joint hypermobility, they should be referred to a genetics clinic for genetic testing and further assessment for HCTD in order to rule out more serious health conditions. In addition, those individuals with joint hypermobility may also benefit from referral to physical and occupational therapists who are familiar with working with hypermobility-related issues, as they may experience greater proprioceptive impairment and poorer overall body awareness [140,141].

## Figures and Tables

**Figure 1 jpm-10-00260-f001:**
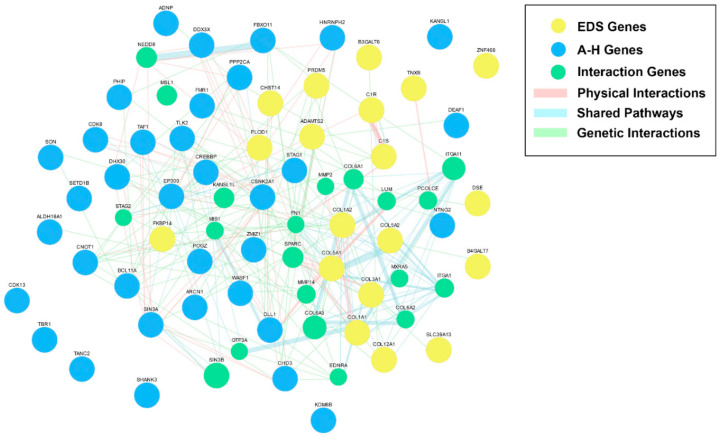
An extended interaction network of genes that are associated with OMIM syndromes comorbid with autism and hypermobility (A–H) (blue) and the Ehlers-Danlos syndromes (EDS) (yellow). An extended interaction network of nodes is shown in green. Direct physical interactions between gene nodes are shown in pink, shared pathways are shown in light blue, and genetic interactions are shown in light green. Note the substantial overlap between A–H and EDS genes, suggesting an extended interactive network and a possible explanation for phenotypic overlap.

**Figure 2 jpm-10-00260-f002:**
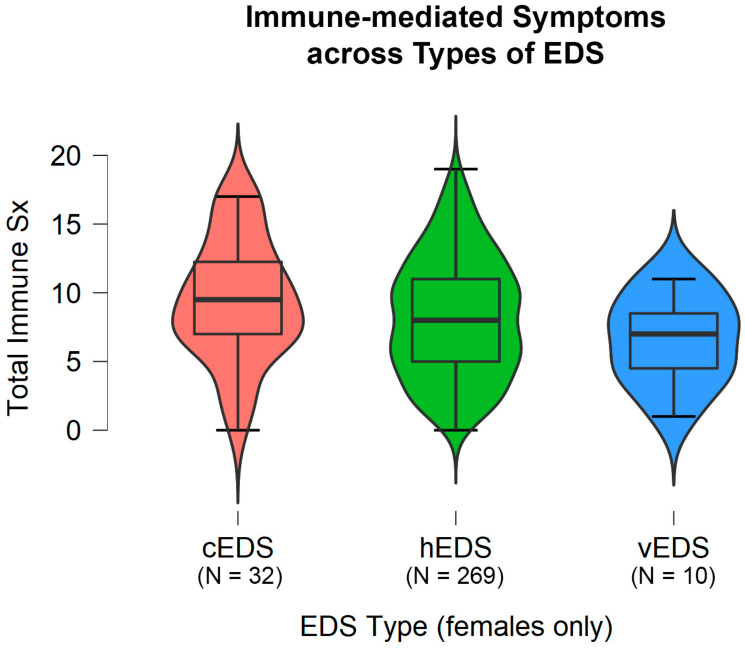
Violin plot indicating the number of reported immune-mediated symptoms across hypermobile EDS (hEDS), classic EDS (cEDS), and vascular EDS (vEDS). Data originally reported in Casanova et al., (2020). Only data for female participants were utilized for the analysis due to low numbers of males in the hEDS group, a condition that is extremely heavily sex skewed towards women.

**Table 1 jpm-10-00260-t001:** Genetic syndromes and their associated genes derived from the Online Mendelian Inheritance in Man (OMIM) database. These syndromes are either associated with autism and hypermobility or the Ehlers-Danlos syndromes. Autosomal dominant = AD; autosomal recessive = AR; X-linked dominant = XLD; X-linked recessive = XLR; unknown inheritance pattern = ?.

OMIM #	Syndrome	Gene/Locus	Inheritance	Group
606053	Intellectual Developmental Disorder with Autism and Speech Delay	*TBR1*	AD	Autism/hypermobility
616603	Cutis Laxa, Autosomal Dominant 3	*ALDH18A1*	AD	Autism/hypermobility
618906	Intellectual Developmental Disorder with Autistic Features and Language Delay, with or without Seizures	*TANC2*	AD	Autism/hypermobility
300624	Fragile X Syndrome	*FMR1*	XLD	Autism/hypermobility
618718	Neurodevelopmental Disorder with Behavioral Abnormalities, Absent Speech, and Hypotonia	*NTNG2*	AR	Autism/hypermobility
610443	Koolen-De Vries Syndrome	*KANSL1*	AD	Autism/hypermobility
615873	Helsmoortel-Van der AA Syndrome	*ADNP*	AD	Autism/hypermobility
615828	Vulto-Van Silfhout-De Vries Syndrome	*DEAF1*	AD	Autism/hypermobility
300958	Intellectual Developmental Disorder, X-linked, Syndromic, Snijders Blok Type	*DDX3X*	XLD, XLR	Autism/hypermobility
618505	Neurodevelopmental Disorder with Coarse Facies and Mild Distal Skeletal Abnormalities	*KDM6B*	AD	Autism/hypermobility
617804	Neurodevelopmental Disorder with Severe Motor Impairment and Absent Language	*DHX30*	AD	Autism/hypermobility
180849	Rubinstein-Taybi Syndrome 1	*CREBBP*	AD	Autism/hypermobility
617140	ZTTK Syndrome	*SON*	AD	Autism/hypermobility
617101	Intellectual Developmental Disorder with Persistence of Fetal Hemoglobin	*BLC11A*	AD	Autism/hypermobility
618354	Neurodevelopmental Disorder and Language Delay with or without Structural Brain Abnormalities	*PPP2CA*	AD	Autism/hypermobility
618205	Snijders Blok-Campeau Syndrome	*CHD3*	AD	Autism/hypermobility
616364	White-Sitton Syndrome	*POGZ*	AD	Autism/hypermobility
613406	Witteveen-Kolk Syndrome	*SIN3A*	AD	Autism/hypermobility
617062	Oku-Chung Neurodevelopmental Syndrome	*CSNK2A1*	AD	Autism/hypermobility
618659	Neurodevelopmental Disorder with Dysmorphic Facies and Dystal Skeletal Anomalies	*ZMIZ1*	AD	Autism/hypermobility
617635	Mental Retardation, Autosomal Dominant 47	*STAG1*	AD	Autism/hypermobility
617991	Chung-Jansen Syndrome	*PHIP*	AD	Autism/hypermobility
618050	Mental Retardation, Autosomal Dominant 57	*TLK2*	AD	Autism/hypermobility
618707	Neurodevelopmental Disorder with Absent Language and Variable Seizures	*WASF1*	AD	Autism/hypermobility
300986	Mental Retardation, X-linked, Syndromic, Bain Type	*HNRNPH2*	XLD	Autism/hypermobility
617164	Short Stature, Rhizomelic, with Microcephaly, Micrognathia, and Developmental Delay	*ARCN1*	AD	Autism/hypermobility
618089	Intellectual Developmental Disorder with Dysmorphic Facies and Behavioral Abnormalities	*FBXO11*	AD	Autism/hypermobility
618709	Neurodevelopmental Disorder with Nonspecific Brain Abnormalities, with or without Seizures	*DLL1*	AD	Autism/hypermobility
619000	Intellectual Developmental Disorder with Seizures and Language Delay	*SETB1B*	?	Autism/hypermobility
617360	Congenital Heart Defects, Dysmorphic Facial Features, and Intellectual Developmental Disorder	*CDK13*	AD	Autism/hypermobility
300966	Mental Retardation, X-linked, Syndromic 33	*TAF1*	XLR	Autism/hypermobility
618748	Intellectual Developmental Disorder with Hypotonia and Behavioral Abnormalities	*CDK8*	AD	Autism/hypermobility
606232	Phelan-McDermid Syndrome	*SHANK3*	AD	Autism/hypermobility
619033	Vissers-Bodmer Syndrome	*CNOT1*	?	Autism/hypermobility
613684	Rubinstein-Taybi Syndrome 2	*EP300*	AD	Autism/hypermobility
130000	Ehlers-Danlos Syndrome, Classic Type 1	*COL5A1*	AD	Ehlers-Danlos syndrome
130010	Ehlers-Danlos Syndrome, Classic Type, 2	*COL5A2*	AD	Ehlers-Danlos syndrome
606408	Ehlers-Danlos Syndrome, Classic-like	*TNXB*	AR	Ehlers-Danlos syndrome
225320	Ehlers-Danlos Syndrome, Cardiac Valvular Type	*COL1A2*	AR	Ehlers-Danlos syndrome
130050	Ehlers-Danlos, Vascular Type	*COL3A1*	AD	Ehlers-Danlos syndrome
130060	Ehlers-Danlos Syndrome, Arthrochalasia Type, 1	*COL1A1*	AD	Ehlers-Danlos syndrome
617821	Ehlers-Danlos Syndrome, Arthrochalasia Type, 2	*COL1A2*	AD	Ehlers-Danlos syndrome
225410	Ehlers-Danlos Syndrome, Dermatosparaxis Type	*ADAMTS2*	AR	Ehlers-Danlos syndrome
225400	Ehlers-Danlos Syndrome, Kyphoscoliotic Type, 1	*PLOD1*	AR	Ehlers-Danlos syndrome
614557	Ehlers-Danlos Syndrome, Kyphoscoliotic Type, 2	*FKBP14*	AR	Ehlers-Danlos syndrome
130070	Ehlers-Danlos Syndrome, Spondylodysplastic Type, 1	*B4GALT7*	AR	Ehlers-Danlos syndrome
615349	Ehlers-Danlos Syndrome Spondylodysplastic Type, 2	*B4GALT6*	AR	Ehlers-Danlos syndrome
613350	Ehlers-Danlos Syndrome, Spondylodysplastic Type, 3	*SLC39A13*	AR	Ehlers-Danlos syndrome
130080	Ehlers-Danlos Syndrome, Periodontal Type, 1	*C1R*	AD	Ehlers-Danlos syndrome
617174	Ehlers-Danlos Syndrome, Periodontal Type, 2	*C1S*	AD	Ehlers-Danlos syndrome
616471	Bethlem Myopathy 2	*COL12A1*	AR	Ehlers-Danlos syndrome
229200	Brittle Cornea Syndrome 1	*ZNF469*	AR	Ehlers-Danlos syndrome
614170	Brittle Cornea Syndrome 2	*PRDM5*	AR	Ehlers-Danlos syndrome
610776	Ehlers-Danlos Syndrome, Musculocontractural Type, 1	*CHST14*	AR	Ehlers-Danlos syndrome
615539	Ehlers-Danlos Syndrome, Musculocontractural Type, 2	*DSE*	AR	Ehlers-Danlos syndrome

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
