# Peer review of "The Relationship between Autism and Ehlers-Danlos Syndromes/Hypermobility Spectrum Disorders"

_jpm, 2020, doi:10.3390/jpm10040260_

Round 1

Reviewer 1 Report

The authors sent a very interesting review summarizing the relationship between autism and Ehlers/Danlos syndromes/hypermobility spectrum disorders (EDS/HSD). The common clinical features connecting both disorders are neurobehavioral, psychiatric and coordination problems, sensory and autonomic dysregulation and immune system disorders. Concerning to the extended interaction network of genes that play a role in both diseases, the authors suggest that EDS/HSD may represent a subtype of autism.

Syndromic autism spectrum disorders represent according to traditional definition a group of childhood neurological conditions, typically associated with chromosomal abnormalities or mutations in a single gene. The discovery of their genetic causes has increased understanding of the molecular pathways critical for normal cognitive and social development.

The submitted review extend our knowledge on similarities between EDS/HSD and autistic spectrum disorders (ASD).

Major points:

  1. Neurological manifestations common in both disorders are described in detail.

However, epilepsy and sleep disorders belong to the main ASD symptoms. It is interesting that epilepsy is a frequent neurological manifestation of EDS, too. The type, the severity and the age at onset of seizures are very variable. Epilepsy represents a frequent cause of morbidity in these syndromes and can influence the long-term evolution of these patients, but the mechanisms are needed to be clarified. The existence of brain lesions has been considered as the main cause.

Similarly sleep disorders ranging from insomnia to obstructive sleep apnea are in EDS very common. The clinical manifestation of EDS includes very frequently also worsened daytime functioning – mainly fatigue and excessive daytime sleepiness. Therefore, narcolepsy or other type of hypersomnia should be excluded as a cause of sleepiness.

These subchapters should be included in the manuscript with adequate references.

  1. Conclusion should be extended; at least the main clinical similarities of both disorders should be mentioned.

Minor points:

Line 24-25: Keywords are missing

Line 68-93: According to my opinion, autism and EDS comorbidity and familial co-occurrence should create an independent chapter

Line 126-129: Supplementary File is missing in my material content

Line 202-208: Should be included in the text

Line 320 and >: Paragraphs containing data on epilepsy and sleep disorders should be added with adequate references

Line 378-379: The sentence is identical with the line 90-91.  

Author Response

We would like to thank the reviewer for his/her excellent recommendations.

MAJOR POINTS:

Neurological manifestations common in both disorders are described in detail. However, epilepsy and sleep disorders belong to the main ASD symptoms. ...

>>>Thank you very much for these excellent recommendations.

A section on epilepsy in EDS/HSD and autism has been added. In addition, we have included a small analysis from previous raw data addressing this issue and included it within this section of the manuscript, finding that EDS/HSD participants reported significantly higher rates of epilepsy than non-clinical sex-matched controls (p = 0.042).

In addition, a section reviewing the literature on sleep disorders in EDS/HSD and autism has been added.

All references for these two additions have been added.

Conclusion should be extended; at least the main clinical similarities of both disorders should be mentioned. 

>>> Clinical similarities have been reviewed more thoroughly and a section has been added concerning precision medicine (recommended by the other reviewer).

MINOR POINTS:

Line 24-25: Keywords are missing

>>> Apologies, keywords have been added to the manuscript.

Line 68-93: According to my opinion, autism and EDS comorbidity and familial co-occurrence should create an independent chapter.

>>> This section has been edited to reflect the requested change.

Line 126-129: Supplementary File is missing in my material content

>>> Apologies, the supplementary file was housed offsite at Mendeley and a link had been included (although mis-placed within the manuscript). This has been corrected. We have opted instead to upload the supplementary file onto the mdpi site and should now be available for download.

Line 202-208: Should be included in the text

>>> It is our understanding that a lengthy quote such as the one we provided needs to be set in an indented paragraph by itself, as opposed to shorter quotes that can be included within the body itself.

Line 320 and >: Paragraphs containing data on epilepsy and sleep disorders should be added with adequate references

>>> These sections and their references have been added.

Line 378-379: The sentence is identical with the line 90-91.  

>>> We have added segues to these two sections so as to reference one another and sound less repetitive.

Reviewer 2 Report

To the authors: This is a well-written, and very intriguing analysis of their analysis of an overlap in symptoms and genetics between autism and Ehlers-Danlos Syndromes/hypermobility spectrum disorders. The review is comprehensive in making their case for the relationship, both using an extensive list of published research studies, and new analyses (i.e. Figure 1). This review would be useful to practitioners, and researchers in both areas.

Suggestions for the authors:

  1. Given the scope of the journal, the authors should make the case for use of personalized medicine in diagnosis and treatment. A separate section on this, or a paragraph in the conclusion would be acceptable to include.
  2. Line 390. Meanwhile: -colon should be removed; font size changes here.
  3. The supplementary files were not supplied for the review. However, it may be useful to consider making these into a table for direct publication with the manuscript, in addition to the full file. Are there overlapping major phenotypes and/or genes that would be better viewed in this format?
  4. Is Figure 2 sufficiently different than the referenced publication to constitute publication, rather than a citation alone? In the online version of (https://jrtdd.com/immune-autonomic-and-endocrine-dysregulation-in-autism-and-ehlers-danlos-syndrome-hypermobility-spectrum-disorders-versus-unaffected-controls/}, as well as a pre-print of the same/similar title available online there appears to have somewhat better information in a comparison of autism to EDS (Figure 1 in published article), compared to Figure 2, which is just EDS subtypes. Can the authors explain the differences, between these two different publications? There is also not a direct overlap of authors. Please also double check the volume/issue number, as according to the journal webpage, the first issue of 2020 is Volume 3, issue 1.

Author Response

We would like to thank the reviewer for his/her excellent recommendations.

Suggestions for the Authors:

Given the scope of the journal, the authors should make the case for use of personalized medicine in diagnosis and treatment. A separate section on this, or a paragraph in the conclusion would be acceptable to include.

>>> A section concerning precision medicine and autism/HCTD has been added to the Conclusions.

Line 390. Meanwhile: -colon should be removed; font size changes here.

>>> We're not exactly sure why these text changes are occurring but may be a result of the website file conversion. It's not in the original .docx manuscript.

The supplementary files were not supplied for the review. However, it may be useful to consider making these into a table for direct publication with the manuscript, in addition to the full file. Are there overlapping major phenotypes and/or genes that would be better viewed in this format?

>>> Apologies, the supplementary file was housed offsite at Mendeley and a link had been included (although mis-placed within the manuscript). This has been corrected. We have opted instead to upload the supplementary file onto the mdpi site and should now be available for download.

Is Figure 2 sufficiently different than the referenced publication to constitute publication, rather than a citation alone? In the online version of (https://jrtdd.com/immune-autonomic-and-endocrine-dysregulation-in-autism-and-ehlers-danlos-syndrome-hypermobility-spectrum-disorders-versus-unaffected-controls/}, as well as a pre-print of the same/similar title available online there appears to have somewhat better information in a comparison of autism to EDS (Figure 1 in published article), compared to Figure 2, which is just EDS subtypes. Can the authors explain the differences, between these two different publications? There is also not a direct overlap of authors. Please also double check the volume/issue number, as according to the journal webpage, the first issue of 2020 is Volume 3, issue 1.

>>> The data in Figure 2 within the current manuscript had been previously published; however, that particular small analysis and the figure are both new. In order to try and clarify this, we have edited this section of the manuscript to make clear we have performed a new analysis on previously published raw data.